# Knowing Is Not Doing: A Qualitative Study of Parental Views on Family Beverage Choice

**DOI:** 10.3390/nu15122665

**Published:** 2023-06-08

**Authors:** Chelsea M. Newman, Jamie Zoellner, Marlene B. Schwartz, Joseph Peña, Kimberly D. Wiseman, Joseph A. Skelton, Tiffany M. Shin, Kristina H. Lewis

**Affiliations:** 1Division of Public Health Sciences, Wake Forest University Health Sciences, Winston Salem, NC 27101, USA; cmnewman@wakehealth.edu (C.M.N.);; 2Department of Public Health Sciences, University of Virginia, Charlottesville, VA 22908, USA; 3Rudd Center for Food Policy & Health, and Department of Human Development and Family Sciences, University of Connecticut, Mansfield, CT 06103, USA; 4Department of Pediatrics, Wake Forest School of Medicine, Winston Salem, NC 27101, USA

**Keywords:** sugar-sweetened beverages, drinking water, beverage, pediatrics, health care, parent, child, tap water

## Abstract

Objective: Sugary drink consumption is associated with adverse health outcomes in children, highlighting the need for scalable family interventions that address barriers to water consumption. To inform development of a scalable, health-care-system-based intervention targeting family beverage choice, a formative qualitative study was conducted using semi-structured interviews with parents whose children were identified as over-consuming sugar-sweetened beverages (SSB) and/or fruit juice (FJ). The first goal of these interviews was to understand, in a diverse real-world patient population, what parents viewed as the primary drivers of their family’s beverage choices, and explore how these drivers might need to be addressed in order to make changes to beverage consumption. A second goal was to explore parental preferences for planned intervention components. An exploratory goal of the interviews was to examine whether knowledge, attitudes, and beliefs around family beverage choice differed across racial and ethnic groups in this sample. Design: Semi-structured phone interviews were conducted and interviews audio-recorded and transcribed. Participants: 39 parents/caregivers of children ages 1–8 who over-consumed sugary drinks as determined by screenings at pediatric visits. Phenomenon of Interest: Parents were interviewed about family beverage choices and preferences to inform development of a multi-component intervention. Analysis: Thematic analysis was performed, including comparison of themes across racial/ethnic groups. Results: Parents expressed that sugary drinks were unhealthy and water was a better alternative. Most were familiar with the health consequences of excess sugar consumption. They identified many reasons why sugary drinks are chosen over water despite this knowledge. One common reason was concern about tap water safety. Few differences were noted across racial and ethnic groups in our sample. Parents were enthusiastic about a technology-based intervention to be delivered through their child’s doctor’s office. Conclusions and Implications: Knowledge is not enough to change behavior. Beverage interventions need to be easy to access, make water more appealing, and elevate beverage choice above the “white noise” of everyday life. Delivering an intervention in a clinical setting could provide an extra level of care, while technology would reduce the amount of live contact and decrease the burden for clinicians and parents.

## 1. Introduction

Sugar-sweetened beverage (SSB) consumption is associated with multiple adverse health outcomes in children and adults, including dental caries, heart disease, and weight gain [1,2,3]. Although U.S. child and adolescent SSB consumption has declined somewhat from a peak of >200 kcal/day in the early 2000’s, beverages still represent the single largest source of added sugars in the American diet [4]. Overconsumption of SSBs remains more common among racial and ethnic minority children and those from lower income households, contributing to ongoing disparities in diet-related health outcomes [2,5]. The reasons for ongoing child overconsumption of SSBs and the persistence of disparities in this behavior are complex, and relate to numerous factors such as industry-targeted SSB marketing strategies, drinking water quality concerns, and the relative low cost and high availability of SSBs.

In light of the ongoing health risks posed by SSB overconsumption, there is a need for scalable interventions acceptable to a diverse population, which address the relevant barriers to healthy beverage choice in families [3,6,7,8,9]. However, traditional behavior change interventions have required large amounts of contact time, making them difficult to translate into busy real world settings, and potentially less accessible to parents of young children. Time and resource demands are particularly salient when developing health-system-based behavioral interventions. Clinical settings introduce many competing priorities, clinician reimbursement for preventive counseling is minimal, and most providers do not have strong training in behavior change counseling [10,11]. Family beverage choice interventions developed for health systems must therefore be tailored to efficiently address the most important barriers to behavior change, and need to be delivered using modalities that do not add substantial burden to the healthcare system.

To inform the development of a scalable, health-care-system-based intervention targeting family beverage choice, semi-structured interviews were conducted with parents whose children were identified as over-consuming SSB and/or fruit juice (FJ). The first goal of these interviews was to understand, in a diverse real-world patient population, what parents viewed as the primary drivers of their family’s beverage choices, and explore how these drivers could be addressed in order to make changes to beverage consumption. A second goal was to explore parental preferences for planned intervention components that would be easy to use and appealing for busy families. An exploratory goal of the interviews was to examine whether knowledge, attitudes, and beliefs around family beverage choice differed across racial and ethnic groups in this sample.

## 2. Materials and Methods

### 2.1. Design and Setting

This was a formative qualitative study using semi-structured phone interviews to capture parental knowledge and beliefs about family beverage choice, understand motivations to change children’s behaviors (or their own/spouse’s behaviors), and explore their preferences for a planned subsequent health care system-based intervention to improve family beverage choices. In a qualitative study such as this one, the goal of the research is to understand the experiences and viewpoints of a group of people on a particular topic. Compared to quantitative research, which often involves large numbers of participants, control groups, statistical analysis, and hypothesis testing, qualitative research is generally geared towards hypothesis generation and more formative work. Therefore, sample sizes are typically much smaller because of the richness of the data that is collected and the lack of statistical testing. For example, a recent systematic review assessing the sample sizes needed for saturation in qualitative research identified a range of 9–17 interviews as an acceptable sample size to reach saturation [12,13]. 

The study took place within the Wake Forest Baptist Health (WFBH) system, a large academic medical center in the Piedmont region of North Carolina. WFBH has a large catchment area, with pediatric primary care practices serving a racially and ethnically diverse patient population encompassing rural, suburban, and urban areas. Since 2017, WFBH pediatric and family medicine practices have been systematically collecting information on self-reported SSB and FJ consumption for pediatric patients ages 6 months through 17 years, using the electronic health record (EHR). These SSB and FJ data are captured in a defined field in a child’s EHR, facilitating easy identification of high-consuming children [14].

### 2.2. Participants

We used the EHR to identify families of children 1–8 years old who met the inclusion criteria for our planned intervention. First, we selected for children who had EHR documentation from an in-person clinic visit in the past month indicating that they currently consumed ≥2 sweet drinks (SSB and/or FJ) per day, and who were documented as English speaking and had parent or caregiver contact information available. Identified potential participants who consumed less than 2 sweet drinks per day, did not speak English, or did not have contact information available were excluded from participation. In light of known racial and ethnic disparities in SSB consumption [2,5], we aimed to over-recruit minority participants relative to our overall patient population. We therefore used a race/ethnicity stratified random sampling technique to generate lists of potentially eligible participants each week, split equally across 3 groups: non-Hispanic Black/African American (NHB), Hispanic, or Non-Hispanic White (NHW) based on the child race and ethnicity documented in the EHR. 

Research staff conducted phone outreach to these children’s parents or caregivers to recruit them and verify eligibility based on EHR data, as well as to verify that they were the primary caregiver of the child in question and willing/able to participate in a telephone interview that would be audio recorded. Those who we were able to reach, and who were interested, were consented verbally, with a printed copy of the consent form then emailed or mailed afterwards, depending on participant preference. Participants were informed that interviews would be recorded for the purpose of transcription and analysis and no identifying information would be reported from the interviews. Participants were compensated with a $25 gift card, provided by mail after completion of the interview. The study protocol and consent process were approved by the WFBH IRB.

### 2.3. Interviewer Training

An experienced qualitative interviewer from the Wake Forest Comprehensive Cancer Center Qualitative and Patient-Reported Outcomes (Q-PRO) Shared Resource trained the study team on qualitative interview techniques. This interactive training lasted several hours and included a background on qualitative interviewing, as well as review of recordings of example interviews and opportunities to practice interview technique. Training topics included developing rapport with participants, how to elicit more information from respondents using techniques such as pauses, probing, and following up or asking clarifying questions, and the practice of writing field notes. Following the training, the first several interviews conducted by each team member were reviewed and team members were provided feedback to further improve interview technique. 

### 2.4. Data Collection

Three trained members of the study team conducted the interviews. Interviews were scheduled at a time convenient for the participant and conducted over the telephone. Before the interview, each participant signed an IRB-approved consent form and was provided a copy for their records. Each interview lasted between 25–30 min and was digitally recorded for transcription. The semi-structured interview guide was developed by the study team and a trained qualitative researcher, and was structured by topics of interest (“domains”) identified by the study team. The first portion of the interview focused on establishing participant understanding of terminology including “sugar sweetened beverage”, “fruit juice”, and “water”. For the purposes of the interview, SSB was defined as any drink with sugars added to it, and FJ was defined as 100% pure fruit juice with no added sugars. This introduction was followed by a discussion of parental SSB knowledge (awareness of health effects, guideline recommendations for child consumption), motivations, facilitators, and barriers to changing family consumption, including an exploration of water consumption. The final portion of the interview assessed parental preferences for planned intervention components (mobile phone application, IVR phone calls, and water bottle toolkit) to inform our intervention design. Before providing feedback, participants listened to a sample IVR call and interviewers described the preliminary mobile phone application design and water toolkit features in detail. Participants were asked to comment on specific components but were also asked to provide open-ended feedback. Finally, participants were offered the opportunity to share any additional ideas or information they had relevant to family beverage choice before finishing the interview. After each interview, interviewers completed field notes summarizing the interview [15], which were reviewed by the lead investigator to ensure the interview guide was eliciting relevant information. All interviews were conducted between March and June 2020.

### 2.5. Survey

Immediately following completion of the semi-structured interview, a predesigned 5 min survey was verbally administered to participants to obtain additional quantitative information about their family’s current beverage consumption habits; their intervention preferences; and demographic information including participant age, family role, educational level, and family structure. Survey responses were collected using the REDCap program. 

### 2.6. Analysis

Interviews were transcribed verbatim, verified by comparing the audio to transcription to ensure accuracy, and de-identified. A modified rapid-analysis approach [16] was utilized to conduct thematic analysis of the interview data [17]. Interview transcripts were reviewed by a trained qualitative research team member and the lead investigator to deductively construct topics of interest (domains) based on the interview guide and inductively identify emerging domains. The qualitative research team member then utilized Microsoft Excel to organize the interview data from all participants, according to domain. Data were then synthesized across participants within each domain, and illustrative quotes were identified. Analytic work by the qualitative research team member was reviewed by the lead investigator for agreement and any discrepancies were discussed and resolved. Finally, data were compared across racial/ethnic groups for differences utilizing a matrix analysis approach [18]. 

Finally, quantitative survey data were exported to a different Microsoft Excel spreadsheet, where we calculated summary statistics [i.e., frequencies for categorical variables, means, and standard deviations (s.d.) for continuous measures].

## 3. Results

### 3.1. Participant Characteristics and Response Rate

Ninety-six potentially eligible parents were reached by phone for recruitment and thirty-nine refused participation or were ineligible upon further screening. While 57 potential participants agreed to complete the interview based on initial phone contact, 18 did not respond at the time of the scheduled interview and were unable to be rescheduled. Semi-structured interviews were completed with 39 participants. Accounting for no-shows, the total response rate for the phone interviews was 41%. 

The demographic characteristics of the sample are presented in Table 1. Of the parents interviewed, the mean (s.d.) age was 31.5 (6.0) years old. All parents interviewed indicated that they were the mother of the child/patient identified in the electronic health record, with the exception of one who indicated they were the father. As shown, there was good distribution across race/ethnic groups and 44% of respondents had a high school education or less.

### 3.2. Participant Terminology

Despite interviewers defining and reviewing beverage types prior to the start of each interview, parents generally did not seem to make any distinction between FJ and SSB when discussing their families’ drink choices. Many appeared to use the terms “juice” and “sugary drink” interchangeably. For simplicity, and to reflect this use of terminology by our participants, we refer collectively to FJ and/or SSB in the results section as “sugary drinks”, unless a participant specifically called out a subtype of beverage.

### 3.3. What Families Are “Supposed” to Drink and Why

The key themes that emerged when we inquired about parental knowledge of what families should be drinking and why they felt that way were that: (1) sugary drinks should be consumed less because they are unhealthy and (2) water should be consumed more because it is healthy (Table 2).

Parents indicated that children and families should drink fewer sugary drinks mainly because of the perceived health consequences of these beverages. They identified conditions such as tooth decay and cavities, weight gain, obesity, diabetes, and high blood pressure as associated with drinking too many sugary drinks. Parents also reported several perceived health benefits of water consumption, such as hydration, healthy skin, cleansing of the body, and increased energy. The factor most parents identified as something that would motivate them to cut back their child’s sugary drink consumption was health—specifically, the prevention of obesity and diabetes.

### 3.4. Reported Sugary Drink Consumption

Despite being selected for an interview based on their child’s over-consumption of SSB or FJ, when asked in the interview whether the amount of sugary drinks their child consumes is “too little”, “just right”, or “too much”, only a third of parents believed the amount was “too much”. However, this varied across racial/ethnic groups. About half of African-American parents believed the amount of sugary drinks their child consumes is “too much”, as opposed to a third of Non-Hispanic White parents, and only two out of the thirteen Hispanic parents.

In the post-interview survey, most parents reported that their children drank sugary drinks at home (74%), while less than half reported that their children drank sugary drinks either at another family member’s house (38%), or at school or afterschool (38%). Parents also reported that child consumption of sugary drinks occurred throughout the day—with 23% reporting consumption during breakfast, 59% with lunch, 46% with dinner, and 49% at snack time.

Parents identified several ways that they could (and sometimes do) go about limiting child sugary drink consumption. The most common strategy reported by about half of the parents interviewed was to restrict access by not buying/bringing sugary drinks into the home. Other approaches mentioned by the parents included: providing more water, making water more appealing by adding fruit, instructing the child to consume sugary drinks in moderation, watering down sugary drinks, setting expectations, and behavior modeling.

### 3.5. Factors Influencing Sugary Drink Consumption

Themes that emerged for *why* families chose sugary drinks were: (1) sugary drinks taste better than water, (2) sugary drinks are more affordable and available than water, (3) sugary drinks are used as rewards for good behavior or to engage kids with meals, and (4) social and/or family influences promote sugary drink intake (Table 2).

When asked to identify reasons why their family consumed sugary drinks, the most commonly reported reason was that the taste of sugary drinks was preferred over water. Parents seemed to report this was especially true for their children, but also occasionally for the parents themselves. The second theme that emerged for facilitators of sugary drink consumption was around availability and affordability. Several parents noted that it was difficult to prevent their children from consuming sugary drinks because these products are ubiquitous—they are available at home, school, fast food restaurants, boxed pre-prepared kids meals (e.g., Lunchables^TM^), and convenience stores. Additionally, one parent noted that in stores, sugary drinks are placed more conveniently and prominently than water, catching their children’s attention. Affordability was similarly mentioned as a facilitator to consuming sugary drinks for some families, with one parent noting that food assistance programs covered the cost of fruit juice.

Regarding the theme that sugary drinks could be a reward or meal incentive, one parent stated that sugary drinks could be used to entice otherwise picky eaters to engage in food consumption. In the post-interview survey, almost half of parents (46%) reported giving sugary drinks to their child as a reward or “treat.”

Finally, a number of social pressures—norms inside and outside the family—were identified by parents as making it more likely that their families would choose to consume sugary drinks. Some parents noted that if children were exposed to sugary drinks by peers or others in their network, it would make it difficult to ask them to drink water instead. For example, they identified the importance of parental role modeling—meaning that the parents themselves were consuming sugary drinks, making it harder to enforce reductions for children.

### 3.6. Barriers to Water Consumption

The majority of barriers to water consumption overlapped with the stated facilitators of SSB consumption reported above, although two additional themes emerged in this section of the interview: (1) water causes unpleasant physical effects, and (2) tap water is unsafe to drink (Table 2).

As noted above, many parents reported that water lacks taste, especially in comparison to sugary drinks, which makes it less appealing to children (and to parents). Additionally, some noted that water may feel less available than sugary drinks (i.e., at a birthday party or while out and about) and in such cases, there are limited options, unless they carried their own water bottles with them. Three parents reported that water causes nausea, heartburn, or indigestion for themself, which makes water unappealing and leads to not setting an example of drinking water for their children.

About half of the interviewed parents expressed concern about the safety of tap water. The look of the water (cloudy or yellow), the smell of the water (suggesting chlorine), the taste of the water (metallic), or nonspecific concerns about “chemicals” in the water contributed to their concerns. We observed differences in perceptions of tap water safety among racial/ethnic groups. Three quarters of Hispanic parents believed their tap water was not safe to drink, while only a third of non-Hispanic Black and non-Hispanic White parents expressed such concerns. The majority of parents who did not trust the safety of their tap water reported buying bottled or filtered water for drinking.

### 3.7. Intervention Preferences

In the final section of the interview, we briefly explored parental preferences for planned intervention components. Representative quotes from this portion of the interview were integrated based on the relevant themes addressed in Table 2. When interviewers described a planned mobile phone application (app) to help families with beverage choices, the vast majority of parents (37 of 39) expressed interest in using this kind of tool to change their family’s behaviors. They reported that their main motivation for using such an app would be to improve their families’ health.

When discussing the planned mobile phone app, parents provided feedback agreeing that the app would be most helpful if it included reminders to track beverages consumed, and an ability to visually display progress or changes in behavior over time. The idea of getting reminders or push notifications from an app produced mixed feelings-some parents felt reminders would be helpful, others were concerned they would become a nuisance. One app suggestion that was provided unprompted by parents was that it would be helpful to see their own and their children’s target water and sugar intake per day. Parents were generally not concerned about the “screen time” aspect of adding another app for their families because they perceived that such an app would be educational and health-promoting.

Parents were also asked to provide feedback on a planned series of interactive voice response (IVR) calls after listening to a sample call about breastfeeding. Most expressed that they would be willing to participate in such IVR calls about family beverage choices and reported they were likely to stay on the phone for the duration of all calls (Table 2). Only four parents interviewed indicated that they would find the IVR calls boring and be unlikely to engage with them. Parents agreed with the topics the study team had identified for potential IVR call content, and agreed that including ‘fun facts’ to keep calls interesting, information about the health consequences of SSB consumption, and tips for parenting through this behavior change would be beneficial.

All of the parents responded positively to the description of a water promotion toolkit distributed by their pediatrician’s office (Table 2). They said they would be likely to use water bottles if provided, and believed a tangible toolkit would help interest their children in drinking more water. Parents agreed with and/or shared their own ideas for additional items to include in the toolkit to enhance its impact on their family, such as stickers for decorating the water bottles, coloring pages, and books about drinking water. Despite high levels of interest in receiving such a toolkit for their family, our post-interview survey showed that most parents (72%) and their children (64%) already had water bottles of their own in the home. Additionally, parents endorsed several concerns about providing children with reusable water bottles, including avoiding bottles with small parts that might pose a choking hazard to toddlers (26%), that children were likely to spill water everywhere (26%), and that they were not sure how to keep the bottle clean (26%).

## 4. Discussion

We interviewed a diverse group of 39 parents to understand their knowledge and beliefs about beverage choices for their children and themselves, and to inform the development of a family-directed intervention embedded in the healthcare system. Consistent with prior research, parents expressed high levels of knowledge and motivation around making healthier drink choices for their families. Adding to the research base, parents identified key barriers that needed to be addressed in order for them to implement changes, including a feeling that changing from sweet drinks to water would be unpalatable and unpleasant for them and their children. To address this difficulty, parents expressed high levels of interest in a convenient, technology-based intervention initiated by their child’s pediatrician.

In part, our findings confirm what has been shown in several prior studies [2,19,20] most parents already know that too much sugary drink consumption is unhealthy for their children and themselves, and that water is a healthier option. Parents were also well-versed in the long-term health consequences that might come from overconsumption of sugars, and endorsed that this was a motivating factor when considering behavior change. However, despite widespread knowledge and motivation, parents indicated a number of barriers to successfully changing behaviors, including children’s dislike of plain water, preferred taste of sugary drinks, availability of sugary drinks, and social or family influences. Additionally, similar to prior studies, our participants frequently mentioned that their families preferred SSBs because they taste better and provide a better drinking experience than water [21,22]. Parents reported choosing SSBs due to their availability and convenience and because many of these drinks come with meals when eating out or as a part of pre-packaged school meals that can be bought straight off the store shelf [19,22]. These factors—enjoyment, availability, and lack of palatable alternatives—highlight the complex reasons as to why parents choose SSBs over water for both themselves and their families, despite knowledge of the health consequences. Our study also supports findings from the existing literature around the topic of water consumption. A recent review article by Patel et al. identified barriers to water intake similar to those that were identified by our sample (boring taste, concerns about tap water, lack of availability) [1].

Unlike some prior studies, ours included a racially and ethnically diverse sample, which allowed us to explore whether barriers to behavior change and potential intervention preferences might differ in a diverse target population. This is particularly important due to known racial and ethnic disparities in sugary drink consumption [2]. There are many cultural and environmental factors that may contribute due to learned behavior from different experiences [23]. While our small sample and methodology preclude making generalizations, one topic where we observed differences according to racial and ethnic group was that of tap water safety, which was an issue more often raised by Hispanic parents. The reasoning provided by Hispanic parents for these safety concerns (worry about water cleanliness and chemical content) was similar to that of NHW and AA parents. Previous literature has identified lower reported tap water and higher bottled water consumption among Hispanic people in the US [1]. This data is congruent with our findings. It is important to note that these differences cannot be attributed to the overarching designation of Hispanic alone, due to the heterogeneity of this population, which has massive cultural differences based on the region of origin (Central America, South America, Caribbean, Spain, etc.) The majority of the Hispanic population within the community that participated in our study are from Central America; therefore, some of these beliefs could represent norms from that region of origin [24].

Unfortunately, the pervasive belief that tap water is unsafe creates a major barrier to water consumption by eliminating a widely available and low-cost source. Yet, tap water is very safe for consumption in the vast majority of U.S. municipalities [1]. Given that the perception of unsafe tap water is also a root cause of high sugary drink consumption, future beverage-related interventions should consider integrating educational content and behavioral change techniques pertaining to the pereception of and access to safe drinking water.

Finally, in preparation for testing a planned health-system-delivered beverage choice intervention, we explored the modality of intervention components, perceived ease of access, and preferences on location for dissemination with participants. Parents were excited about the proposed intervention, which was to include a video, mobile phone application, and automated educational and motivational calls. They were especially supportive of the idea of initiating this intervention by having their child’s pediatrician office provide them with a water promotion toolkit (water bottles, stickers, and a children’s book), citing that this step showed an extra level of care by the pediatrician. Most prior family beverage interventions have taken place in other settings (i.e., schools, home, community centers, etc.) rather than in clinical or healthcare settings [3,7], making a pediatrics-delivered family beverage choice intervention somewhat unique.

Additionally, most prior interventions have required a large amount of live contact with interventionists (either by phone or in person) [3], which could place a burden on already-busy parents and severely limit the generalizability and sustainability of these interventions outside of controlled clinical trial settings. Currently published systematic reviews highlight the need for scalable interventions that are acceptable to diverse populations [3,6,7,8,9]. Our hope, supported by these interviews, is that an intervention with a strong focus on water promotion for the whole family (as opposed to just warning parents about children’s sugary drink intake), and featuring a warm hand-off by the pediatrician may better address the barriers to healthy drink choice expressed by parents in our sample.

The use of technology to deliver most aspects of the intervention may be a good way to thread the needle of requiring multiple contacts for behavior change, while not placing substantial additional burden on the healthcare system or parents of young children. The idea of remote, technology-based interventions for beverage choice has previously been successfully tested by Zoellner et al., who found IVR phone calls to be an effective component of a beverage choice intervention in rural Appalachian adults [25]. However, whether these findings will translate to a health-system-delivered intervention for families, without additional face-to-face support for parents, is unknown. Finally, while some research supports use of water bottle giveaways [2] and mobile apps [6] individually as interventions, we are unaware of any similar prior interventions in the pediatric clinical setting that combine all of these modalities without requiring a high frequency of in-person contact [1,6]. Combining these modalities should be explored to determine the potential to increase reach to racial/ethnic minorities.

### 4.1. Limitations

Several study limitations should be noted. First, participants came from a small geographic area in the Southeastern U.S.; therefore, our findings may not be generalizable to all populations. Likewise, since interviews were conducted in English, our sample of Hispanic parents may not represent the views of those who were excluded due to speaking Spanish only, and which could have revealed different viewpoints. Second, the timing of the interviews coincided with the onset of the COVID-19 pandemic. Although we had hoped to use parental focus groups to facilitate rich discussion around these topics and provide more hands-on interaction opportunity for testing intervention components, the protocol had to be revised to collect data using telephone interviews instead, due to an inability to conduct research data collection in person. This meant that participants could not see examples of the intervention options when they were providing feedback. Additional limitations include the lack of data related to certain social or demographic characteristics of the participants, including household income. However, self-reported education level is a good proxy and tends to yield more complete data [26]. These potential limitations should be considered within the context of the study strengths, including a sampling strategy that yielded a diverse sample, interviews conducted by a trained research team and guided by a semi-structured guide, a robust and iterative thematic analysis approach, and an adequate sample size to achieve data saturation.

### 4.2. Conclusions

Overall, our interviews support the idea that parents know what their families should drink and why, but that knowledge is not enough to consistently change behavior. It is clear that beverage interventions for busy families need to be easy to access; make water more fun, tasty, and appealing; and regularly prompt and remind parents to focus on the issue, elevating beverage choice above the “white noise” of everyday life. Interventions with these features may be more likely to succeed than those that require substantial additional time commitment or that focus primarily on correcting knowledge deficits. Our findings also support prior research suggesting that beverage interventions need to include an educational component around tap water safety. This feature is especially important if interventions are to succeed in reducing racial and ethnic disparities in sugary drink consumption. Parents liked the idea of an intervention being delivered through their child’s pediatrician’s office, suggesting that delivering an intervention in the clinical setting could be valuable and effective. Future work should assess the impact, if any, of socioeconomic status versus cultural norms on SSB and water consumption. Following further development of the intervention components and refinement of the behavioral content to address the emergent themes reported here, our team will conduct a clinical trial of the intervention components delivered by a child’s pediatrician’s office. Currently, the planned intervention components include a water bottle toolkit to be distributed by the child’s pediatrician, an IVR call series, and access to a mobile phone application.

## Figures and Tables

**Table 1 nutrients-15-02665-t001:** Descriptive Characteristics of Interview Participants.

Characteristic	N/% for Categorical Variables, Mean (sd) for Continuous Variables
Female Sex	38 (97%)
Age (years)	31.5 (6.0)
Race ^a^	
White	18 (46%)
Black or African American	11 (28%)
Other or Prefer not to Say	10 (26%)
Ethnicity ^a^	
Hispanic	13 (33%)
Non-Hispanic	24 (62%)
Other or Prefer not to Say	2 (5%)
Marital Status	
Married	20 (51%)
Single	16 (41%)
Separated	3 (8%)
Educational Level	
Less than high school	3 (8%)
High school or GED	14 (36%)
Some College or 2-year Degree	17 (44%)
4-year college degree or higher	3 (8%)
Prefer not to Say	2 (5%)

^a^-obtained based upon self report, and reflective of parental race and ethnicity, which may or may not have aligned with the electronic health record documented race and ethnicity of the child used for sampling.

**Table 2 nutrients-15-02665-t002:** Themes and Representative Quotes from Parental Interviews, By Interview Topic.

Domain	Theme	Quotes from Participants
What Families are “Supposed” to Drink, and Why	Sugary drinks should be consumed less because they are unhealthy	General knowledge and beliefs: “What I heard before form my doctor is pretty much saying that it’s not as good. Especially instead of doing sodas and juices like Kool-Aid, do water....” (011-HISP) “The way I was raised was it’s terrible for you. Stay away from it. It’s no good for your teeth. It’s no good for your body...” (2010-NHW) “That in excess, it may have adverse effects on their health and wellness, weight gain, diabetes, their access to other issues regarding their health.” (002-AA) “That sugary drinks are not good for you… sugary drinks are not healthy, so different types of medical problems, high blood pressure, diabetes, things that they associate it with.” (025-AA) Intervention preferences relevant to this theme: App: “People with diabetes and stuff like that can’t eat, drink sugary stuff. I just wanna know what kind of drinks out there. Some suggestions, ‘cause that will help me out a lot, and I could change my whole grocery list up.” (2014-NHW)
	Water should be consumed more because it is healthy	General knowledge and beliefs: “...Water is life... It’s no other chemical in there. Water is clear. It won’t mess you up. Water is like, it will clean your insides out.” (015-HISP) “Pretty much all my life, people just say it’s good for you and your body ’cause you need water to really survive so you don’t get dehydrated.” (014-AA) “That it’s essential. It’s recommended for better health and management for them.” (002-AA) “It’s really healthy for them to drink water, helps with keeping them from being dehydrated, and several other parts of their body, staying at the full function and that’s about it.” (010-NHW) Intervention preferences relevant to this theme: App: “It would be nice to sit and have a reminder tell me, ‘Hey, it’s time for you to drink a bottle of water. You really need one.’ That’s gonna stop me in my tracks and help me stop and say, ‘Okay.” (2010-NHW) Toolkit: “It would be very helpful. I’d actually enjoy the fact that my doctor’s actually trying to help with my child’s health as far as at home and not just her doctor’s visits.” (014-NHW)
Factors Influencing Sugary Drink Consumption	Sugary drinks taste better than water	General knowledge and beliefs: “The fact that it tastes good. It’s got flavor to it. Tastes like juice. It goes down easier. It’s sweet.” (028-NHW) “I honestly don’t know why it’s so easy for the kids to pick a sugary drink, versus just grabbin’ a cup and gettin’ some water. I think because sugary drinks are—they have color to them, and they have a flavor to it. Kids like stuff that has a flavor too.” (026-AA) “[*water*] has no flavor to it, so you get bored. You wanna have a taste in your mouth, and so teas or any other sugary drinks are more desirable for the taste palate, for us at least.” (022-NHW) “The preference for me, I like the water taste better, but my kids like—they like the sugary drinks. When they’re drinking water—it’s a little bit harder to get them to drink the water or juice. They want a soda. They want that bubbly, that fizz in their mouth type sugary drink.” (2003-NHW) Intervention Preferences Related to this Theme: Toolkit: “I would gladly accept. I would do anything to try to keep on the water path….It’s just finding something that can keep stimulating it, to drink it. Being able to infuse it with different fruits to get a different flavor every day, that would help. (025-AA)
	Sugary drinks are more affordable and available than water	General knowledge and beliefs: “Usually it’s more cheaper than it would be regular bottled water, and it would be more convenient like at the gas station, like if we go down the road to get a soda versus a water, because of the cost.” (022-NHW) “Well, one, it comes on my WIC. The fruit juice. I get ‘em for free. They do drink a lot of those, and usually they’re cheaper than buying the waters and stuff from the store.” (019-AA) “Oh, my kids get more of their sugary content from school, so during school, the chocolate milk more so. I do know that they give—I think they give—the juice I think is 100% juice, but the chocolate milk… and then my older kids, because they have access to vending machines and stuff like that, so they can get their major sugar access from there.” (025-AA) “It’s convenience. It’s not often, but it’s convenience. If we don’t have nothin’ to drink in the refrigerator, it’s easier. I literally live right behind [*gas station*] ***, across from the hospital in [*town*] ***. It’s easy just to run to [*gas station*] ***, grab two two-liters.” (2010-NHW)
	Sugary drinks are used as rewards for good behavior or to engage kids with meals	General knowledge and beliefs: “My son, if he wants a soda, he has to pick up his room to get a soda, ‘cause they are not allowed to have a bunch of sodas.” (027-NHW) “Well, particularly my daughters, they’re really picky eaters, and having something good to drink… they’ll be more interested in eating a pasta… they’re picky, I mean, picky. So I say, “You guys can have a little bit of soda,” and they get excited.” (005-HISP) “Yeah, I would rather have a soda to go with my french fries and burger, instead of some water.” (2011-AA) Intervention preferences relevant to this theme: Toolkit: “ I will be interested in knowing how it looks and how they did it and I think it will be nice. Also, it’s motivating because the kids, when they see something new or like a new cup and stuff like that, they feel more motivated. They just want to drink from that. It would be interesting.” (2012-HISP)”
	Social and/or family influences promote sugary drink intake, including competing priorities as a busy parent	General knowledge and beliefs: “For my kids, it makes it harder for them to drink water when they see me not drinking water too.” (001-HISP) “Well if we’re not at home and we’re at someone else’s house, or if we’re somewhere like at a birthday party or something. We went to birthday parties at a park before with kids. Parents bring…we haven’t took our own drink to have. They’ll drink whatever is there, available for them to wash down their food with.” (024-NHW) “I guess sometimes when they see other kids drinking other things, they don’t wanna drink water. That may make it a little harder. They wanna drink what the other kids are drinkin’.” (030-NHW) “It’s easy to let what we drink become white sound in the background for the other things that go on” (022-NHW) Intervention preferences relevant to this theme: App: “That’s the biggest thing, is I think the app would give him an incentive. It shows him he has something to work towards, to do better, and you get rewarded for it.” (024-NHW) App (push notifications): “Maybe like once a day, or give you the ability to change it. ‘Cause I know initially we need more constant reminders, where as it became second nature and I used it more frequently, I wouldn’t need as many reminders.” (003-AA) IVR calls: “I guess it would have to make sure it states that this is what it’s for, in relation to the drink process, make sure it state that within the first fifteen seconds that we’re on the phone. That way I know, ’cause with everything going on, I’m getting a whole lot of spam calls, so a lot of times, when you hear automated voice, you’re like, “Oh, this is a scam, gonna hang up now.” (2011-AA) IVR calls: “Yes, I would probably stay on the phone, and it would probably change some of the behaviors and some of the drink choices that I would probably make for the rest of that day.” (022-NHW)
Barriers to Water Consumption	Water causes unpleasant physical effects	General knowledge and beliefs: “I know with myself, it causes more heartburn and indigestion to drink water.” (010- NHW) “I’m not a big fan of water… It makes my stomach hurt.” (2001-NHW)
	Tap water is unsafe to drink	General knowledge and beliefs: “I believe they should drink filtered water or bottled water, because I believe it’s cleaner than tap water would be.” (001-HISP) “I personally give my children bottled water. I feel like it’s better, more safer than just the tap.” (023-HISP)

* Identifying information was removed.

## Data Availability

Data are unavailable for sharing due to privacy reasons—participants did not provide consent for data to be shared outside of the study.

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
