# Peer review of "Knowing Is Not Doing: A Qualitative Study of Parental Views on Family Beverage Choice"

_nutrients, 2023, doi:10.3390/nu15122665_

Round 1

Reviewer 1 Report (Previous Reviewer 2)

I appreciate the answers to my previous comments. I see the text's improvement however information on interventions is still missing. I understand that the Authors present this topic in another paper, but just a sentence or two about the issue would improve the general understanding of their research concept. 

No other remarks.

Author Response

Thank you for pointing this out. Additional information about the planned intervention components has been added. For convenience, this is pasted below:

Currently, the planned intervention components include a water bottle toolkit to be distributed by the child’s pediatrician, an IVR call series, and access to a mobile phone application.

Reviewer 2 Report (Previous Reviewer 1)

Reviewer’s results for the article of Nutrients 2338769 V2

Major comments

 I would check according to my previous reviewing comments and their responses of the authors. My comments are written in black and the authors’ responses are in blue:

 This is the article to show the study of parental knowledge is not enough to change their children drinking behaviors for health. The follows are my concerns to this article:

1.        Before the authors start interviewing to parents, they conducted interview training. This seems scientific and important for validating study method. However, this is not controlled study and the authors did not take control group that has no training.

No response by the authors in the revised version.

2.        In table 1, annual income must be included in parental characteristics because it must impact on eating and drinking behaviors such as eating and drinking frequencies and impact on drinking amount of sugar.

No additional information of annual income in table 1 added by the authors in the revised version.

3.        The numbers of subjects must be too small to draw conclusive results.

No response by the authors in the revised version, especially in the limitations of this artcile.

As almost my comments seem ignored.

However, the article contents seem scientific and relevant to the readers.

As the result of my second reviewing, my conclusion for this article is accepted.

Author Response

Thank you for taking the time to review this resubmission. We appreciate your feedback and have tried to provide clear responses. Please see the attachment for full responses.

Reviewer 3 Report (Previous Reviewer 3)

To inform development of a scalable, health care system-based intervention targeting family beverage choice, the authors conducted semi-structured interviews with parents whose children were identified as over-consuming SSB and/or fruit juice.
There is no need to mention the contribution of each author in the article. You have separate paragraph for that.
Any study requires the informed consent of the individuals and considerations of ethical issues, and these must be approved by the Ethics Commettee. You have not mentioned anything about this information în the article. Please add this information în the text.
The article presents 26 references, being up to date. However, I recommend that you add other references as well.
It is a clear, correct article, but in my opinion it presents predictable results, without bringing new information.

Author Response

Thank you for your feedback. We have tried to address your concerns adequately. Please see the attachment for our full responses.

This manuscript is a resubmission of an earlier submission. The following is a list of the peer review reports and author responses from that submission.

Round 1

Reviewer 1 Report

Reviewer’s results fro the article of Nutrients 2338769 V1

Major comments

 This is the article to show the study of parental knowledge is not enough to change their children drinking behaviors for health. The follows are my concerns to this article:

1.        Before the authors start interviewing to parents, they conducted interview training. This seems scientific and important for validating study method. However, this is not controlled study and the authors did not take control group that has no training.

2.        In table 1, annual income must be included in parental characteristics because it must impact on eating and drinking behaviors such as eating and drinking frequencies and impact on drinking amount of sugar.

3.        The reason to have to choose sugar drinking shown in table 2 seems hard to understand for readers. The authors might made mistake to choose method to show their results of interviews.

4.        The numbers of subjects must be too small to draw conclusive results.

In general, the reason to choose sugar beverages for children must be important for their health because it might have an impact on long-term health issue. However, the methods in this study have many limitations.

As the result, my conclusion for this article is major revision.

Minor comments

1.        No information about body sizes of parents and children.

2.        No information of annual income for each subjects.

3.        Too small number of subjects.

Reviewer 2 Report

The paper covers an interesting topic. Unfortunately, it isn't easy to assess its contribution to science. 

First of all, we have different objectives mentioned in the paper.  The first one was related to understanding what parents viewed as the primary drivers of their family’s beverage choices, and exploring how these drivers might need to be addressed in order to make changes to beverage consumption. I would say that it was partially met as very general proposals were prepared in terms of addressing the drivers for more healthy beverage consumption. 

The second one was about exploring parental preferences for planned intervention components. As the authors do not present the interview's scenario, we do not know whether these components were just ideas of respondents or maybe they were presented by the interviewers and interviewees just commented on them. The lack of the scenario's description is problematic in terms of judging the obtained results as well.

The last objective was about examining whether knowledge, attitudes, and beliefs around family beverage choice differed across racial and ethnic groups in this sample. This objective was met.

None of the objectives is presented in the abstract so after reading just the abstract, one does not know what is the purpose of the paper.

The presentation of the results could be better in terms of its clarity and catching the attention of the reader. A very long table, with many quotations, without highlighting the most important phrases or words, is hard to keep the attention of the reader. There are more "catchy" ways of presenting the results of a qualitative study than the table. I would strongly suggest working on this issue.

The last thing that needs to be improved is the topic of the interventions. It is not presented in a way that proves the authors' knowledge in this specific area. I would recommend expanding the analyses about the types of interventions, their objectives, and areas of influence.

I hope that the authors will find my comments useful in terms of improving the paper. Good luck!

Reviewer 3 Report

This is a formative qualitative study using semi-structured phone interviews to capture parental knowledge and beliefs about family beverage choice.
Line 36... use the  same parenthesis for references '"(1) (2, 3)"
You have the paragraph with "Author Contributions" , so it is no longer necessary to specify the contribution of each author in the methodology.
You have not specified the criteria for excluding participants. Please add this information.
The article presents 23 references. In reference 22 you did not specify when you accessed the link (day, month, year).
In my opinion, the study does not bring new information and the number of participants is too small (only 39 participants) it should be enlarged. The harmful effect of sugar and ignorant behavior on the consumption of sugary foods and drinks are known. 

Round 2

Reviewer 1 Report

Reviewer’s results for the article of Nutrients 2338769 V2

Major comments

 I would check according to my previous reviewing comments and their responses of the authors. My comments are written in black and the authors’ responses are in blue:

 This is the article to show the study of parental knowledge is not enough to change their children drinking behaviors for health. The follows are my concerns to this article:

1.        Before the authors start interviewing to parents, they conducted interview training. This seems scientific and important for validating study method. However, this is not controlled study and the authors did not take control group that has no training.

No response by the authors in the revised version.

2.        In table 1, annual income must be included in parental characteristics because it must impact on eating and drinking behaviors such as eating and drinking frequencies and impact on drinking amount of sugar.

No additional information of annual income in table 1 added by the authors in the revised version.

3.        The numbers of subjects must be too small to draw conclusive results.

No response by the authors in the revised version, especially in the limitations of this artcile.

As almost my comments seem ignored.

As the result of my second reviewing, my conclusion for this article is reject.

Reviewer 3 Report

the authors answered my comments

Author Response

Thank you for your review and comments which have improved the manuscript.